# Organize, Don’t Agonize: Strategic Success of *Phytophthora* Species

**DOI:** 10.3390/microorganisms8060917

**Published:** 2020-06-17

**Authors:** Jane Chepsergon, Thabiso E. Motaung, Daniel Bellieny-Rabelo, Lucy Novungayo Moleleki

**Affiliations:** 1Department of Biochemistry, Genetics and Microbiology, Forestry and Agricultural Biotechnology Institute, University of Pretoria, Pretoria 0002, South Africa; jane.chepsergon@up.ac.za (J.C.); thabiso.motaung@up.ac.za (T.E.M.); 2Department of Ecology and Environmental Sciences, Umeå University, 90187 Umeå, Sweden; daniel.bellieny@gmail.com

**Keywords:** effectors, *Phytophthora*, hemibiotroph, oomycete

## Abstract

Plants are constantly challenged by various environmental stressors ranging from abiotic—sunlight, elevated temperatures, drought, and nutrient deficits, to biotic factors—microbial pathogens and insect pests. These not only affect the quality of harvest but also the yield, leading to substantial annual crop losses, worldwide. Although plants have a multi-layered immune system, phytopathogens such as species of the oomycete genus *Phytophthora*, can employ elaborate mechanisms to breach this defense. For the last two decades, researchers have focused on the co-evolution between *Phytophthora* and interacting hosts to decouple the mechanisms governing their molecular associations. This has provided a comprehensive understanding of the pathobiology of plants affected by oomycetes. Ultimately, this is important for the development of strategies to sustainably improve agricultural production. Therefore, this paper discusses the present-day state of knowledge of the strategic mode of operation employed by species of *Phytophthora* for successful infection. Specifically, we consider motility, attachment, and host cell wall degradation used by these pathogenic species to obtain nutrients from their host. Also discussed is an array of effector types from apoplastic (hydrolytic proteins, protease inhibitors, elicitins) to cytoplastic (RxLRs, named after Arginine-any amino acid-Leucine-Arginine consensus sequence and CRNs, for CRinkling and Necrosis), which upon liberation can subvert the immune response and promote diseases in plants.

## 1. Introduction

The challenges of food security are extensive and can surpass efforts to increase food production. Biotic factors such as plant weeds, insect pests and pathogens are estimated to cause over 20% of losses in food crops, and hence are recognizable players in crop losses that consequently place a long-term burden on the global economy [1,2]. Of the three biotic factors, phytopathogens are the main threat to crop health and natural ecosystems, and several of them can impose overwhelming consequences to the quality of harvest if not well managed [3].

The genus *Phytophthora* was classified under the kingdom fungi for a long time partly due to morphological similarities between the two groups [4]. Currently it is classified as oomycetes as phylogenetic revealed that oomycetes are stramenopiles together with algae specifically diatoms and brown algae [5], as summarized in Table 1.

Presently, more than 150 *Phytophthora* species are recognized, and additional species are being identified yearly [10], totaling 10 clades that are phylogenetically known [11]. Some species are able to infect several plants and are hence termed “broad range”, while others have a narrow host range (Table 2).

A survey conducted by Kamoun et al. [27] reports the top 10 oomycete pathogens with economic importance and impacts to food security and natural ecosystems preservation globally. Among these are six *Phytophthora* species which feature *P. infestans* (the potato blight pathogen) at the top of the list. *Phytophthora* can cause up to 100% of loses in many plant species annually and billions of dollars are spent every year to mitigate associated diseases however, little success has been recorded [11]. Therefore, pathogens of this genus certainly live up to the name *Phytophthora* that literally translates as “plant destroyers”, a term coined by a German scientist [28] when he was describing the potato late blight pathogen [29]. Indeed, this warrants a systematic analyses and understanding of key steps that these pathogens take to infect their hosts.

Over the years, researchers have spent a lot of time exploring the driving forces underpinning the success of pathogenic *Phytophthora* species. A large number of studies have been dedicated to the relationship between the biology of species belonging to this genus and plants they infect [4,16,30,31,32,33]. Some of the useful findings of this concerted effort relate to the hemibiotrophic lifestyle of *Phytophthora* species [16]. This parasitic lifestyle requires a living host cell during the early phase of infection and subsequently transition to a devastating necrotrophic phase, which is epitomized by the secretion of effector proteins that contribute to the capacity of pathogens to uphold the biotrophic phase for some time and to also survive post biotrophy. Therefore, hemibiotrophic pathogens such as members of *Phytophthora* can be understood better through studying this lifestyle and how it facilitates host invasion and subsequent death.

To date, recent advances in the understanding of *Phytophthora* pathobiology have focused on the molecular basis of virulent cytoplasmic and apoplastic effectors that contribute to disease [34,35,36,37,38]. In addition, emphasis has been made on where in the host cell cytoplasmic and apoplastic effectors are translocated, as well as their host interacting partners, to subvert the plant immune system. To put these into perspective, the present review broadly discusses the infection strategies of *Phytophthora spp* and highlights how these pathogens respond to plant exudates, as well as attach and acquire nutrients from the host plant. Also discussed in detail are the effectors used during these strategies.

## 2. Strategy I: Declare War on the Enemy

In the presence of sufficient water in the soil (free water) and on leaf surfaces, *Phytophthora* species begin the infection process by swimming to the target host using biflagellate zoospores and thereafter attach to the plant using adhesive proteins [32] as demonstrated in Figure 1.

### 2.1. Drawing Near to the Enemy: Motility of Zoospores

Zoospores of *Phytophthora* species are asexual, and were labelled by Judelson and Blanco [4] as “the weapons” of the plant destroyer. They are wall-less, uninucleate and biflagellated chemotactic cells aiding the pathogen both in reaching the plant and locating optimal infection sites [4,39]. With the aid of the flagella, these pathogens swim for up to 24 h in free water, followed by rapid encystment [40]. Taurocyamine kinases in *P. infestans* zoospores have been shown to help storing energy as well as shuttling high energy phosphoryls from mitochondria to flagella [32].

It is worth mentioning that zoospores detect gradients of specific compounds like sugars, amino acids and ions originating from host roots. In a chemotactical and electrotactical manner, they can be attracted towards the exudates leading to host specificity, in both root pathogens like *P. capsici, P. parasitica* and *P. sojae* [41] and foliar pathogens [33,42]. For instance, zoospores of *P. sojae* tend to respond to isoflavones [43] while those of *P. infestans* are attracted to amino acids such as glutamine [42]. Associated guanidine-binding (G) protein subunits, alpha (Gα), beta (Gβ), and gamma (Gγ) are known to regulate extracellular and intracellular signaling pathways in eukaryotes [44]. These proteins have also been implicated in regulating how zoospores of oomycetes respond to host plant exudates [45]. Analysis of G proteins showed that the knocking down of alpha G-protein subunit inhibited chemotaxis of *P. infestans* zoospores [46]. A specific group of G-proteins named membrane-bound G-protein-coupled receptors (GPCRs) have been reported to contain a C-terminal phospholipid kinase domain in most *Phytophthora spp* [44]. Consequently, mutation of this domain in *P. infestans* yielded defects in motility of zoospores [45]. Histidine triad nucleotide-binding protein 1 (PsHint1) in *P. sojae* was involved in zoospore chemotaxis, cyst germination and pathogen’s virulence [46]. Similarly, *P. sojae* Gα protein (PsGPA1) was reported to be essential for zoospore functioning since mutation of the gene led to defects in both zoospore chemotaxis and encystment [47]. Recently, PsGPA1 was shown to negatively regulate sporangium formation by suppressing the kinase activity of threonine protein (PsYPK1) [48].

Collectively, zoospore motility is an essential process that is catalyzed by G proteins and therefore, the focus now should be on the downstream pathways or proteins facilitated by these G proteins as well as the signaling established by the single G protein complex.

### 2.2. Stick, Stick and Stick: Attachment of Zoospores

After successful zoospore motility, a further facet to the success of *Phytophthora* is the capacity to adhere and establish within host tissues as biotrophs (Figure 1). Beyond motility, these pathogens immediately bend their ventral grooves towards the host to release thrombospondin protein in the direction of the host plant surface before encystment [30]. The pathogen cells attach firmly to the outside of the potential host leading to cell entrance and further enhances the development of disease structures, which is a requirement for penetration [30]. While working on *P. nicotianae*, Zhang et al. [49] identified a conserved protein containing short consensus repeats (10 kDa) with a beta-sandwich like arrangement called the Sushi domain. This domain lands onto the plant’s surface from other zoospore vesicles by “kiss and run” exocytosis mechanism leading to cell-cell adhesion. Oomycete adhesion proteins, most of which comprise of mucin and jacalin-like proteins, cellulose-binding elicitor lectin (CBEL) as well as acidic cell wall proteins [50,51], protect germinating cyst against desiccation [33]. The contribution of CBEL to binding hyphae to cellulosic substances was experimentally validated through knockdown assays in *P. parasitica var-nicotianae* [50]. A study conducted on *P. cinnamon* revealed that small dorsal vesicles of the pathogen secrete large glycoproteins (>330 kDa) from a mucilage-like coating on the surface of the cyst [17]. Similarly, the germinating cyst of *P. parasitica* can secrete mucin-like (MUCL) proteins onto the host surface [51]. Since mucins form a highly hydrated barrier that act against pathogen invasion in animals [16], it is therefore hypothesized that mucins secreted by the zoospores and germinated cysts of the above mentioned *Phytophthora* species serve similar protective functions during the infection process.

## 3. Strategy II: Turn the Tables by Counteracting!

Following successful attachment of the pathogen to the host, penetration into the plant apoplast is paramount for successful infection. The plant apoplast is well known as a battlefield between plants and invading pathogens [52]. As the pathogen progresses to invade the host cell, the plant in turn responds to the pathogen by producing catalytic classes of proteases that prevent further invasion by the attacker [53]. To counteract this, the pathogen can release inhibitors of those enzymes. A hallmark of apoplastic effectors, just as the name suggest, is their interaction and outcome that takes place outside the plant host cell membrane. To this effect, *Phytophthora* species secrete a wide range of apoplastic effectors including cell wall-degrading enzymes (CWDEs), enzyme inhibitors and elicitins [37].

### 3.1. Tearing Down a Complex: Cell Wall Degrading Enzymes

Following adherence of *Phytophthora* to the host plant cell wall, the degradation of this physical barrier will then ensue. Owing to the complexity of the cell wall, the successful *Phytophthora* species may be forced to activate the secretion of CWDEs that specifically target hemicelluloses, cellulose, pectins, β-1,3-glucans and glycoproteins, thereby reducing the complexity of the cell wall structure and successfully gaining entry and colonizing the host [33]. Blackman et al [54] showed that *Phytophthora* species express more pectin targeting CDWE than fungi, and have predicted 423 and 431 such proteins likely to be secreted CWDEs by *P. infestans and P. parasitica* respectively. In a follow up study, those researchers carried out a transcriptomic analysis of 200 CWDEs in *P. parasitica* where pectinases, hemicellulases, cellulases and β-1,3-glucanases were reported to be highly expressed during biotrophic phase of infection [55]. A separate study predicted a total of 696 genes encoding CWDEs in *P. cactorum* with 282 predicted to be potentially secreted from the pathogen during infection [4,56].

Commonly reported CWDEs in species of *Phytophthora* include glyceraldehyde hydrolases, carbohydrate binding molecules, carbohydrate esterase, pectin lyase, and glycosyl transferases [16]. Interestingly, CWDEs targeting β-1,3-glucan were predicted to be the main cell wall components of *Phytophthora* that act as microbe-associated molecular patterns (MAMPs) activating the plant’s first-line of defenses [57]. However, as was later noted by Armitage et al. [56], the presence of β-1,3-glucanase in *Phytophthora* species could be functioning in callose breakdown.

According to research performed on *P. infestans,* the apoplastic effector *in planta*-induced protein (IPI-O) can affect the cell wall of the host. Key to this is the tripeptide Arg-Gly-Asp (RGD) motif of the effector which interrupts the integrity of the plant’s cell wall through lectin receptor kinase LecRK-I.9 binding [33]. Another apoplastic effector, PE1 with pectin-binding domain, was shown to localize around the haustoria of *P. infestans* [58]. This effector was associated with early infection of the potato plant by enhancing the formation haustoria leading to a close contact between the pathogen and the host cell. Based on these studies, it is clear that members of *Phytophthora* have means to interfere with the integrity of the host cell wall, eventually gaining entry into the host.

### 3.2. Deny the Enemy Targets: Secretion of Protease Inhibitors

After bringing down a citadel, the pathogen is now presented to the apoplast. Here, the conserved MAMPs are perceived by the host plant recognition receptors (PRRs) leading to the activation of the host’s first line of defense, commonly known as pattern-triggered immunity (PTI). Examples of MAMPs in *Phytophthora spp* that have been well characterized include: the elicitin infestin 1 (INF1), Pep13 and cellulose binding elicitor lectin (CBEL) [33,50]. One of the events that are activated in the PTI phase includes the delivery of proteases into the apoplastic space with a plan to degrade effectors secreted by *Phytophthora* [59]. To counteract the host defenses, the pathogen secretes apoplastic effectors, mainly protease inhibitors. For instance, *P. infestans* secretes cystatin-like cysteine protease inhibitors EPIC1-EPIC4 and EPIC2B (Figure 1) to inhibit defence responses of *Phytophthora* Inhibited Protease 1 (PIP1) in the tomato plant [60]. Similarly, three cysteine protease inhibitor genes were predicted in the wide host range pathogen *P. cactorum* [56]. Recently, a total of 80 cysteine proteases in *P. parasitica* was identified and further characterization was done on PpCys44 and PpCys45 shown to trigger cell death in various species of *Nicotiana* [61].

Another interesting group of inhibitors is the serine proteases that prevent degradation of the pathogen cell wall component [62], and several key examples will be discussed. Extracellular serine protease inhibitor (PpEPI 10) from *P. palmivora* is secreted to counter defenses deployed by *Hevea brasiliensis* serine protease (HbSPA) [63]. In a recent study [64], the apoplastic effector PsAvh240 of *P. sojae* was demonstrably able to interact with host secreted aspartic protease GmAP1 in the plant plasma membrane, blocking its entry into the apoplast and leading to suppressed soybean immunity. Another *P. sojae* endogenous apoplastic effector named PsXEG1 can be inhibited by soybean glucanase GmGIP1, although the pathogen can circumvent this inhibition by secreting PsXLP1 effector with inactive enzymatic activity [65]. PsXLP1 can bind more tightly to GmGIP1 than PsXEG1, leading to the release of PsXEG1 to promote virulence of *P. sojae* on soybean [65].

Selective pressures seem to define the *Phytophthora* protease-inhibitors’ ability to thwart activity of proteases of their specific host plants. For instance, the extracellular cystatin-like protease inhibitor EpiC1 of *P. infestans* was shown to exhibit an inhibition specificity that differs from that of its ortholog PmepiC1 of *P. mirabilis* (Figure 1). This was attributed to the specificity to changes in one amino acid in PmepiC1 and its substrate, Mirabilis Rcr3-like protease 2 (MRP2) with Asp^152^ as a key element of specificity. Thus, these two effectors are categorized into their respective hosts based on the specificity of protease-inhibitor to their protease partners [59].

### 3.3. Small Bites While Attacking: Secretion of Elicitins

Still in the apoplastic space, *Phytophthora* species secrete a group of small and structurally conserved proteins called elicitins. These proteins have been shown to have no sequence similarity to plant proteins hence they are classified as MAMPs. Most oomycetes secrete both α-class (acidic) and the β-class (hydrophillic residue) elicitins [37]. Examples include capsicein in *P. capsici*, cinnamomin in *P. cinnamomi*, cryptogein in *P. cryptogea* and parasiticein in *P. parasitica* [37].

Elicitins are highly expressed during *Phytophthora*-host interaction causing cell surface recognition that trigger an immune response [66,67]. For a long time, elicitin in most oomycetes including *Phytophthora* have been known to induce hypersensitive reaction (HR) cell death in host plants [68,69]. Nonetheless, host specificity in response to *Phytophthora* elicitins has been reported [70]. Furthermore, elicitins induce HR cell death in specific host plants [71].

Of note, species of *Phytophthora* are sterol auxotrophs, meaning they are unable to synthesize sterol, hence acquiring it from the environment. As such, elicitins are supposed to act as sterol-carrier proteins in facilitating acquisition of sterols [58]. It was hypothesized that elicitins could induce cell death by interfering with the integrity of the plasma membrane during sterol binding [70]. Nonetheless, in previous studies, elicitin mutants of *P. infestans* failed to bind sterols from potato plants but rather, elicited cell death, signifying an insignificant relationship between sterol binding and cell death response [72,73]. It is therefore evident that elicitins in the genus *Phytophthora* are key players in sterol binding during infection. As sterols and fatty acids stimulate sexual reproduction and particularly oospore production [74], one can speculate that most elicitins in *Phytophthora* species could contribute to interspecies variation, potentially giving rise to more virulent strains. This is so because sexual oospores allow for genetic recombination leading to genetic variability.

## 4. Strategy III: The Inner-Front Ploy

At the plant-pathogen interface, an effector is secreted by a pathogen then translocated to a potential host cell, making the host environment beneficial to the pathogen [34].

### 4.1. Destroy from Within: Cytoplasmic Effectors of Phytophthora Species

Cytoplasmic effectors (RxLR and CRNs) of oomycete pathogens are trafficked in to the host cell where they are directed to different subcellular organelles [75]. RxLRs are typified by the presence of signal peptide (SP) at the *N*-terminal region which facilitates their secretion through the conventional pathway. The SP is then followed by an RxLR-EER motif and a variable C-terminal region or effector domain [76,77]. An exemplary RXLR effector in oomycetes has a highly conserved Arg-X-Leu-Arg (with X being any amino acid) or the RXLR motif located within 32 amino acids of the SP. After the RXLR motif is the EER motif, denoting Ser/Asp-Glu-Glu-Arg [78]. This motif bears similarity to translocation signals in *Plasmodium* species and maybe involved in trafficking proteins into the host [75]. On the other hand, the CRNs (CRinkling and Necrosis) possess a conserved Leu-Phe-Leu-Ala-Lys (LFLAK) motif that is found within the first 60 aa at the *N*-termini (<130aa) [79]. The motif is implicated in translocation of the effector into host cells [80]. To date, genomes of several *Phytophthora* species have been sequenced with *P. infestans and P. litchii* showing the largest and lowest genome sizes of 240 and 38 Mb, respectively. In addition, RxLR and CRN effectors in various genomes of *Phytopthora* species have been predicted where no correlation between genome size and the number of effectors predicted is observed (Table 3).

The C-terminal region of RxLRs contains the WY motif of tryptophan (W) and tyrosine (Y) residues, concealed in the hydrophobic core of the helical roll, and is found in about 44% of *Phytophthora* RxLRs [88,89]. Although variation in WY motif has been recorded in most RxLR effectors of *Phytophthora* species [35], modification of the W and Y residues has no significant effect on protein folding provided the hydrophobic potential is preserved [77]. Unlike the RxLRs, the C-domain of CRNs consists of subdomains as seen in *P. infestans*, having 36 different conserved subdomains of C-terminal subfamilies [79]. In *P. capsici*, a total of seven new specie-specific C-terminal subfamilies was recorded, which suggests CRN domain expansion in *Phytophthora* [81]. What this expansion entails for the life cycle of these pathogenic species is yet to be clear, although it is anticipated to contribute to specific pathogenic lifestyles [81].

### 4.2. The End Justifies the Means: Translocation of Cytoplasmic Effectors

Cytoplasmic effectors display an effective role in host invasion [73]. The SP targets the effector to the endoplasmic reticulum (ER) where it is cleaved to permit secretion into the extra haustorial matrix [35]. Indeed, this has been reported for oomycete effector delivery with the RxLR signal modulating effector trafficking into the host cells [36,75]. Furthermore, this SP-dependent translocation process appears to be independent of the pathogen’s machinery [36,75]. In line with this, effectors of human malaria parasite *Plasmodium falciparum* revealed an RxLX motif that is crucial for effector delivery into the host cytoplasm [90].

Following much work on RxLR effector delivery and translocation, Dou et al. [91] reported that both the RxLR and Asp-Glu-Glu-Arg (DEER) motifs are important for proper host targeting in most effectors, although according to Whisson et al. [75] the RxLR motif alone may be sufficient. This notion stemmed from the fact that modifying the RxLR motif to four alanines (AAAA) prevented the delivery of the effector [75]. A molecular mechanism that supports RxLR-mediated host entry was later proposed in *P. sojae* effectors PsAVR1b, PsAvh331 and PsAvh5 [92]. This mechanism is explained by the RxLR motif binding to the phospholipid, precisely phosphatidylinositol-3-phosphate (PI3P), on the plasma membrane of the host cell, which is then transferred into the cell through lipid raft-mediated endocytosis. A careful analysis of this molecular mechanism further demonstrated that the translocation function of the RxLR motif can be affected by amino acid polymorphisms in the RxLR effectors [e.g., RXLR to Arg-any aa-Phe-Leu-Arg (RFLR) → Phe-Arg-Leu-Arg (FRLR) or RFLR → RFRL], and this can lead to deficient phospholipid effector binding [92]. With this, *Phytophthora* RxLRs would be present in the vesicles derived from the host plasma membrane [92,93]. Although this delivery mechanism has been highly accepted by many, it was challenged through experiments involving the RxLR domains of Avr3a from *P. infestans* and Avr1b from *P. sojae* [78]. This later study provided new evidence that the RxLR domain of *P. infestans* (Avr3a and Avr1b) alone was insufficient to facilitate entry into the host plant, contrary to what was initially proposed [75]. The study was soon supported by other studies showing that RXLR motifs are indeed not enough for binding to PI3P [93]. The C-terminal domain was in fact a new player in effector-phospholipid binding, as well as the virulence role in the host plant cell [78,93].

A further modification to the canonical way of *Phytophthora* RxLR effector translocation was proposed in 2017, but this time, biochemical studies of the well characterized RxLR effector AVR3a from *P. infestans* were carried out [94]. These studies pointed to the fact that the RxLR motif of the effector is stabilized by acetylation followed by cleaving prior to secretion. More recent work on *P. infestans* revealed that the RxLR effector Pi04314 is secreted through a non-conventional pathway [58]. The pathway seems to involve direct shuttling of SP-containing proteins from the ER to the plasma membrane by passing the Golgi, hence the name Golgi-bypass or type IV pathway [95]. Combined, these studies have enhanced our understanding of RxLR effector delivery and translocation in *Phytophthora* despite the lack of a consensus on how RxLR effectors of oomycetes are translocated into plants.

Similar to RXLR motif, LFLAK motif of CRN effectors has been implicated in effector translocation as reviewed by Amaro et al. [96]. For instance, the *N*-terminal LFLAK motif of CRNAVR3a in *P. capsici* was changed to Leu-Ala (LAAAA) leading to infection of the host by this oomycete [80]. The authors concluded that although the Leu-X-Leu-Phe-Leu-Ala-Lys (LXLFLAK) domain is not found at the *N*-terminal region of all CRNs, the motif is nonetheless important in delivering CRN into the host cytoplasm. For both CRN and RxLR effectors of *Phytophthora*, effector activity could reside in the C-terminal regions of the protein, as previously reported [88], where RxLR effectors have an abundance of short α-helices at the C-terminus important to effector functional adaptation. Similarly, the C-terminus of *Phytophthora* CRN is responsible for effector function [81]. Altogether, the precise function of LFLAK motif in effector protein delivery into host cell remains vague just as the RxLR-dependent translocation mechanism of *Phytophthora* effectors.

### 4.3. After the Trigger is Pulled: Subcellular Localization of Effectors

With the help of sorting or transit signals, cytoplasmic effectors traffic to distinct organelles and associate with plant proteins that are key in immune response [80]. All *Phytophthora* CRNs that have been identified localize to the host nucleus while RxLRs enhance host colonization by localizing to different organelles of the host cytoplasm including the nucleus [97]. Significant associations between the localization of cytoplasmic effectors and their virulence function have been reported [80,97,98].

Most RxLRs in species of *Phytophthora* have been shown to traffic to the various compartments of the cell like the nucleus, cytosol, plasma membrane or ER [99]. These include the well-studied RxLR effector AVR3a of *P. infestans*, which often localize to the cytoplasm and nucleoplasm, the RxLR effectors of *P. infestans*, Pi04314, Pi03192, PexRD54 and PexRD18 which either localize to (or associate with) the nucleus and nucleolus, endoplasmic reticulum, autophagosomes and plasma membrane, respectively [98,100,101,102].

Assessment of 52 RxLRs in *P. infestans* indicates that most of these effectors localize to the ER, mitochondria, peroxisomes, or microtubules [99]. Recently, *P. infestans* effector Avrblb2 was shown to localize to the plasma membrane where it interacts with the host target [103]. In a separate study, *P. capsici* effector RxLR48 co-localized to the plant nucleus together with its host target Nonexpressor of Pathogenesis-Related Proteins 1 (NPR1) [104]. Another effector of *P. capsici,* PcAvr3a12, co-localized with its host target protein to the endoplasmic reticulum [105].

It is worth mentioning that some of *Phytophthora* effectors interact with their targets and relocate to the action site where they effectively execute their function [106]. PiAVR3a^KI^ from *P. infestans* is a nucleo-cytoplasmic effector operating in this fashion; when it is expressed together with its host targetin *N. benthamiana*, it is re-trafficked to late endosomes with the target [107]. Similarly, elicitin inhibiting RxLR effector Pi02860 localizes to the nucleus and its exclusion from this site is able to inhibit INF1-triggered cell death [108]. This confirms that the nucleus is not the prime site of Pi02860 host cell performance. In a recent study [106], *P. sojae* effector PsAvh52 was found to localize to the cytoplasm although its action site is the nucleus. Despite this relocalization of *Phytophthora* effectors, a countable number of phytopathogen effectors have been shown to cause host target re-trafficking actions [98].

By contrast, some of RxLRs have been shown to localize and carry out their activity at the same host organelle. For instance, nucleolus and nucleus are native compartments of *P. infestans* effectors Pi04314 and Pi04089, respectively, and could not accomplish their activity outside of these compartments [100,109]. In summary, *Phytophthora* RxLR effectors traffic to diverse sites of the host cell with these sites playing a significant association with the pathogen’s virulence activity.

### 4.4. Hit Where it Hurts Most: Effectors Target Key Components of Host Immunity

One mechanism that *Phytophthora* species employ to suppress the host response is targeting crucial defense proteins [110]. Studies show that most RXLRs of *Phytophthora* target various events of PTI to successfully complete the biotrophic phase. For instance, mitogen-activated protein kinase (MAPK) cascades are crucial for host defense signaling [111]. However, RxLR effectors of the well-studied *P. infestans* (PexRD2, SFI1 and SFI5) interact with the kinase domain of MAPKs to subdue host response [112,113,114,115]. Regulation of reactive oxygen species (ROS) is another important event of PTI that is targeted by pathogens. This is true in RxLR effector activity of *P. capsici* (RxLR48) and *P. sojae* (Avr3b), which inhibit ROS-mediated defense responses, enhancing pathogen colonization in their target hosts [104,116]. Another significant facet of plant defense is the phytohormone-associated signaling (Salicylic acid (SA), jasmonic acid (JA), ethylene (ET), and auxin) [117]. Nonetheless, plant pathogens hijack these pathways to promote disease. One way to achieve this is through the use of effectors, such as in the case of the RxLR effectors Pi04314 from *P. infestans*, PsIsc1 and Avh238 from *P. sojae* and RxLR48 from *P. capsici.* These effectors employ a common *modus operandi* by suppressing JA and SA hormonal levels and also involving an interplay with certain metabolic pathways important for generating the precursors of these hormones [100,104,118]. Other plant steroids like brassinosteroids (BRs) are involved in crosstalk with defense signaling pathways [119]. However, RxLRs from *P. infestans* (PiAVR2 and PexRD2) have been shown to impede the BRs kinase 1 (BAK1)-Cf4/AVR4 dependent cell death (Figure 2) to promote disease progression [120,121].

Apart from targeting positive regulators of plant immunity, recent studies have revealed that successful *Phytophthora* effectors target plant proteins called susceptibility factors (SFs). These factors critically promote compatible host-pathogen interactions [100]. Various *P. infestans* and *P. sojae* RxLRs have been reported to target SFs [109,122,123]. Although SFs are gradually coming to light, how *Phytophthora* species exploit these factors to suppress immunity remains elusive. A few studies have demonstrated that RXLR effectors target SFs interacting with positive regulators of immunity for proteasome-mediated degradation [100,108,124] as seen in Figure 2.

Autophagy is also an important process implicated in stress tolerance and defense against pathogens through the build-up of defense hormones and HR to prevent the spread of microbial infection [125]. However, effectors of *Phytophthora* species can hijack this process [103]. For instance, the PexRD54 from *P. infestans* is known to interact with an associate of the autophagy-related (ATG) protein (ATG8) [89,102,126,127]. Another *P. infestans* RxLR effector AVR1 interacted with sec5, an exocyst component of the potato plant leading to cell death suppression [67]. *P. brassicae* effector RxLR24 interacted with various Rab guanosine triphosphate phosphatases (GTPases) to inhibit host vesicular trafficking [128]. Recently, AVH195 of *P. parasitica* was found to impede autophagy process by interacting with autophagy-related protein ATG8 leading to reduced autophagic flux while favoring pathogen proliferation [129].

As indicated in Table 4, there are other important processes that different plant hosts clearly require for survival. Nonetheless, *Phytophthora* species through the use of various effectors target these processes to enhance disease development.

## 5. Strategy IV: The Fait Accompli

Cytoplasmic effectors (CRNs and RxLRs) occupy the same battle ground but employ different approaches to manipulate the host immunity. As previously described, CRNs and RxLRs promote and suppress PTI, respectively [81]. It is also hypothesized that the two effector types play a role in biotrophic (RxLRs) and necrotrophic stage (CRNs) of oomycete hemibiotrophy [81]. After successful completion of the biotrophic phase, the pathogen transition to the cell killing phase where they secrete CRNs before exiting the dead host to infect fresh plants through sporulation. Nonetheless, since some *Phytophthora* species live on perennial hosts for a long time like a chronic disease, it is therefore worth mentioning that not all *Phytophthora* species kill their host at the time of sporulation.

### 5.1. The Final Combat: CRN Effectors Induce Cell Death

CRinkling and Necrosis effectors (CRNs) in *P. infestans* were first observed to cause necrosis when expressed ectopically in plants [134]. Since then, a series of reports have supported this concept. Although it is believed that *Phytophthora* species kill their host after successful completion of the biotrophic phase [33], it is not clear whether all species of *Phytophthora* employ this mechanism before exiting to infect fresh plants through sporulation.

One study analyzing the CRN effector domain of PiCRN8 from *P. infestans* identified kinase activity involving auto-phosphorylation during expression *in planta* [135]. However, a kinase mutant of PiCRN8 is associated with reduced necrosis [135], suggesting this CRN effector can induce cell death. This seems to be a widely conserved function of CRN effectors as others in this group (e.g., CRN20_624, CRN79_188, CRN83_152 and CRN4 in *P. capsici*) also induce cell death upon ectopic over-expression *in planta* [81,136]. Interestingly, some effectors (e.g., CRN20_624) may have an additive effect on the PTI interface by inducing PAMP induced cell death [81].

Despite the clear association with cell death, some of CRN effectors in *Phytophthora* do not conform to this rule. These have been reported to suppress host cell death processes, as demonstrated in *P. sojae* where cell death induced by elicitors is suppressed [137]. PsCRN63 and PsCRN115 induces and suppresses programmed cell death, respectively, although the underlying mechanism remains unclear [138]. However, some groups have begun elucidating the mechanisms, elaborating that PsCRN63 and PsCRN115 of *P. sojae* interact directly with, and cause relocation of catalases from the peroxisomes to the nucleus [139]. To date, we understand that PsCRN63 may promote plant cell death by down-regulating catalase stability while up-regulating hydrogen peroxide levels, while PsCRN115 counters the functions of PsCRN63 [139]. Furthermore, PsCRN63 acts by suppressing immunity responses induced by flg22 such as callose deposition [140]. Cross-talk between CRN effectors may be a widely occurring phenomenon. For instance, PpCRN7 and PpCRN20 in *P. parasitica* enhance and suppress INF-induced cell death in *N. benthamiana*, respectively [85]. Regardless of this cross-talk in relation to cell death, the two effectors still augment susceptibility of *N. benthamiana* to the pathogen. Taken together, CRN effectors are important for *Phytophthora* species as cell death regulators *in planta*, but so is cross-talk which may be critical to select which effectors are more effective at what stage of host cell death. Nonetheless, the implication of CRN-induced cell death remains a matter of conjecture.

### 5.2. Exiting the Battle Field: Sporulation in Phytophthora Species

*Phytophthora* species have been reported to develop very fast, giving little time for agronomists to counter their effects on time [141]. This could be attributed to the polycyclic nature of the disease where a large number of asexual spores are produced and dispersed leading to successive infections through sporulation. Sporulation in *Phytophthora* is influenced by various environmental factors like availability of nutrients, humidity, amount of oxygen and pH [14]. Furthermore, at night in *P. infestans*, this process is temperature-sensitive and favored by high humidity [142]. This explains why sporangia of *P. infestans* are more prone to midday desiccation since they lack pigments for blocking ultraviolet light [33,143]. Sporulation determinants may differ within *Phytophthora* species. For instance, starvation can induce sporulation in *P. sojae* but not in *P. infestans* [142].

Studying the molecular basis of sporulation could enhance our understanding of the process and reflect on the differences and infection potential of different *Phytophthora* species. Recent studies revealed some mechanisms that may be key in regulating sporulation in species such as *P. infestans*. For instance, the downregulation of nitrogen metabolite repression regulator at onset of sporulation is accompanied by the upregulation of the catalase gene PiCAT2 during asexual reproduction and late infection stage [5,11,32]. This suggests PiCAT2 is important for the sporulation process. Indeed, PiCAT2 has been confirmed to be indispensable for both the formation and function of sporangia of the potato late blight pathogen [144]. Sporulation in oomycete pathogens can also be stimulated by viruses (e.g., PiRV-2), as viral nucleic acids can induce transcriptome changes which favour spore development in oomycetes [141]. The study further postulated that PiRV-2 could stimulate sporulation through restriction of ammonium and amino acid intake [141]. This is of particular importance given that several oomycetes (and fungi) are known to harbor viruses [145]. However, the principal cause for sporulation in *Phytophthora* remains to be defined [33].

## 6. Conclusions

There is no doubt that studies on phytopathogen infection including those caused by *Phytophthora* species are focused on tackling how these phytopathogens access their potential host and most importantly, how do they suppress host immunity for disease development. We have therefore discussed four strategies that *Phytophthora* species employ for successful infection. One of these strategies is the ability of cytoplasmic effectors to target not only the positive regulators of plant immunity but also the negative regulators or the susceptibility factors for enhanced disease progression. It would be now important to investigate whether these effectors are widely conserved among species of *Phytophthora* and whether they target plant proteins which are also conserved, commonly known as “immune hubs”. This could be an avenue of potential value to crop protection and enhanced food production as it could be leveraged by plant breeders by engineering durable and broad-spectrum resistance. Taken together, understanding how *Phytophthora* species manipulate the host is paramount to device innovative strategies to effectively manage their destructive diseases.

## Figures and Tables

**Figure 1 microorganisms-08-00917-f001:**
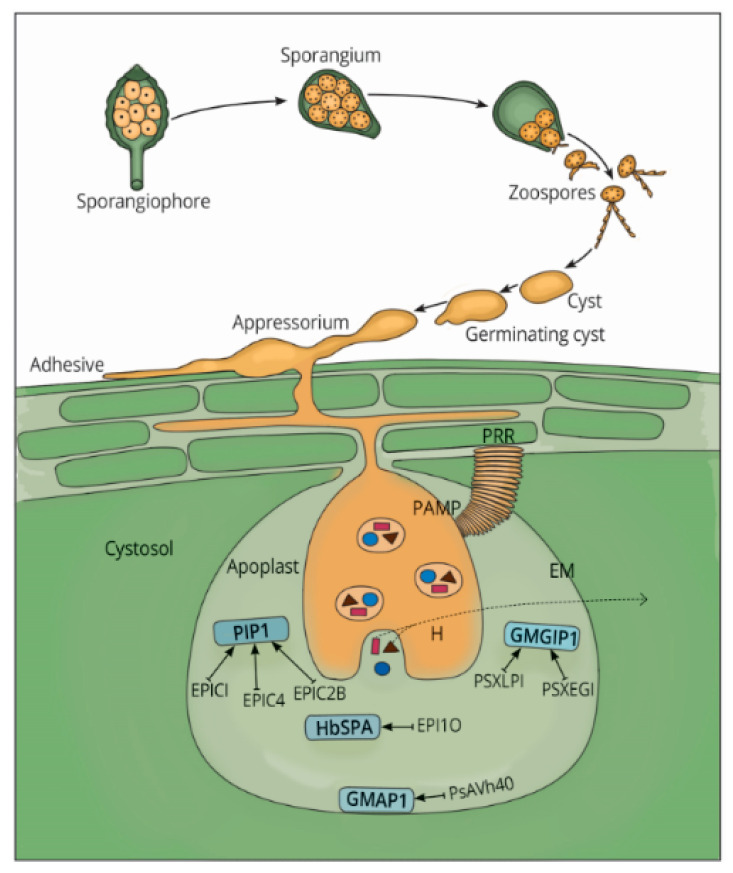
Proposed model of *Phytophthora* species infection process and the apoplastic effectors inhibiting plant proteases. The dispersal phase starts with multinucleate sporangium which releases zoospores which encyst on the plant host then germinate to swelling at the end of the germ tube which attach to the host cell using adhesive. Thereafter, it penetrates the host cell using the appressorium to form intercellular hyphae which is commonly known as infection vesicle that grows between host cells to form a haustorium (H) that invaginates the host cell membrane (EM) to secrete apoplatic effectors (in blue) RxLRs (purple) and CRNs (brown). Apoplastic effectors inhibit (→) key plant protease in bold; *Phytophthora* Inhibited Protease (PIP1), *Hevea brasiliensis* serine protease (HbSPA), soybean aspartic protease(GMAP1) and soybean glucanase (GMGIP1). The conserved PAMP is perceived by the host plant recognition receptor (PRR) leading to the activation of the host’s first line of defense.

**Figure 2 microorganisms-08-00917-f002:**
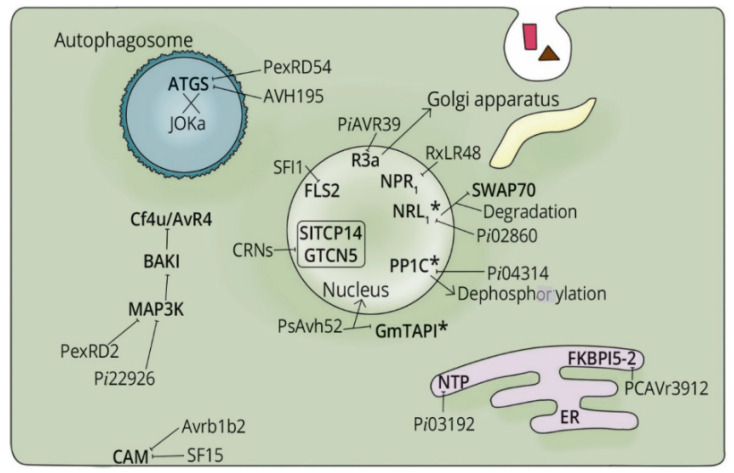
Proposed model of *Phytophthora* cytoplasmic effectors localizing to different subcellular compartments as they act on various plant proteins that are key in immune system. Most RxLR effectors target positive regulators of the plant immunity while a few of them target negative regulators/susceptibility factors in asterisks (*). The arrow (→) denotes effector subcellular relocalization, while (⊣) designates effector-target interaction.

**Table 1 microorganisms-08-00917-t001:** Differences between true fungi and oomycetes.

Feature	Oomycete “Fungi”	True Fungi	References
**Biochemical/cytological**			
Thallus	Aseptated mycelium	Septated mycelium	[4]
Ploidy of hyphae	Diploid except for temporary gametangia haploid nuclei	Usually haploid with semi-stable diploid stage after mating	[4]
Cell-wall component	Cellulose and β-glucans	chitin, (N-acetylglucosamine residus 1,4-linked (1,4-GlcNac))	[1,6]
Sterol synthesis pathway	Absent-obtain-sterol from host-sterol carrier proteins-during infection	Present	[7,8]
Flagellated asexual spores	Biflagellated zoospores	Absent-except for chytrids which are monoflagellate	[4]
**Molecular**			
Neighboring taxonomic group	Brown algae, diatoms, and Apicomplexans	Animals	[9]
Typical genome size	50–250 Mb	8–177Mb	[4]

**Table 2 microorganisms-08-00917-t002:** *Phytophthora* pathogens with their respective diseases and hosts.

*Phytophthora* Pathogen	Disease	Host	Reference
*P. alni*	Root and collar rot	Alders	[12]
*P. brassicae*	Rot in store	Brassicaceae	[13]
*P. cactorum*	Foliar leaf spots, shoot blight, root and crown rot	Wide-host range	[14]
*P. capsici*	Leaf blight, stem and fruit rot	Wide-host range	[15]
*P. cinnamomi*	Root rot and root crown rot	Wide-host range	[16,17]
*P. citricola* (currently *P. plurivora*)	Dieback of trees, shoot blight, root crown rot, and root rot	Wide-host range	[18]
*P. fragariae*	Red core disease	Strawberry	[19]
*P. ilicis*	Foliar leaf spots, shoot blight, and stem cankers	Holly (*Ilex*)	[14]
*P. infestans*	Leaf late blight	Potato, tomato and *Solanum spp*	[20]
*P. lateralis*	Root and collar lesions	Cupressaceae family (cedar and cypress)	[21]
*P. litchi*	Downy blight	Longan, litchin species	[22]
*P. meadii*	Root rot and leaf fall	Citrus, cocoa and black pepper	[23]
*P. megakarya*	Pod rot	Colanut, cocoa	[2,14]
*P. melonis*	Root and fruit rot	Cucurbits	[14]
*P. palmivora*	Leaf blight, pod, bud and fruit rot	Wide-host range	[14]
*P. parasitica*	Root and stem rot	Wide-host range	[14]
*P. phaseoli*	Downy mildew	Lima bean	[14]
*P. ramorum*	Foliar leaf spots and shoot blight, Bleeding stem cankers in Oaks	Wide-host range	[24]
*P. sojae*	Root and stem rot	Soybean, lupin	[25]
*P. syringae*	Foliar leaf spots, shoot blight, and stem cankers.	Wide-host range	[3,26]

**Table 3 microorganisms-08-00917-t003:** Predicted number of RxLR and CRN effectors in Phytophthora species.

*Phytophthora spp*	Genome Size (Mb)	RxLR	CRN	References
*P. cactorum*	121.5	199	77	[56]
*P. capsici*	64	357	84	[15,81]
*P. cinnamomi*	58	171	45	[16,79]
*P. infestans*	240	563	450	[79]
*P. litchii*	38	245	14	[82]
*P. megakarya*	126.8	336	152	[83]
*P. multivora*	41	84	60	[4,10,84]
*P. palmivora*	151.2	415	137	[83]
*P. parasitica*	64.5	172	80	[85,86]
*P. ramorum*	65	350	60	[81,87]
*P. sojae*	95	350	202	[81,87]

**Table 4 microorganisms-08-00917-t004:** Host cytoplasmic effector targets in *Phytophthora* species.

Effector	Origin	Host Target	Function	Reference
PsAvh262	*P. sojae*	BiPs	PsAvh262 to stabilize endoplasmic reticulum (ER)-luminal-binding immunoglobulin proteins (BiPs), resulting in attenuated plant defense responses	[130]
Avrblb2	*P. infestans*	CAM	Avrblb2 interact with calmodulin (CAM) interfering with plant defense associated Ca^2+^ signaling in plants	[103]
CRN12_997	*P. capsici*	SlTCP14–2,	CRN12-997 subvert host immunity by targeting SlTCP14-2 leading to its mislocalization	[81]
PsCRN108	*P. sojae*	Heat Shock Protein (HSP)	PsCRN108 associates with heat shock elements (HSEs) hence suppressing its expression	[131]
RxLR48	*P. capsici*	NPR1	The effector interacts with NPR1 (non-expressor of pathogenesis related-1), which functions as the central signaling regulator during systemic acquired resistance leading to PTI suppression	[104]
PSR1; PSR2	*P. sojae*	PINP1	PSR1 binds to RNA helicase (PINP1) interfering of miRNAs and siRNAs PSR2 reduces the accumulation of siRNAs in extracellular vesicles and subdues the conserved gene-silencing machinery	[132,133]
PcAvh1	*P. capsici*	PP2Aa	PcAvh1 associates with the protein phosphatase PP2Aa, a key component of plant immunity	[110]

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
