# Peer review of "Organize, Don’t Agonize: Strategic Success of Phytophthora Species"

_microorganisms, 2020, doi:10.3390/microorganisms8060917_

Round 1

Reviewer 1 Report

In this review, molecular mechanisms of plant infection published for Phytophthora species are summarized.  The MS is quite well organized and informative.  However, wrong information and careless mistakes are scattered over the manuscript.  All authors are encouraged to read references more in-depth and also verify every gene and symbol in Figures and Tables carefully.

Specific comments:

In Table 2, soybean is one word.  P. sojae can infect several Lupinus species, but soybean is the only economically important host.  This indicates that “host range” can be artificial.

Figure 1.  PIP1 not described in the text.

GmGIP1, GmAP1: gene names should be consistent.

EPIC2B, missing a letter.  Shouldn’t it be EpiC4?

Directions of suppression arrows in Fig. 1 are all wrong.  For instance, the arrow indicates as if HbSPA suppresses EPI10, but it is the other way around.  Please also enlarge arrowheads to make them easier to see.

Line 86 asexual, abd were (typo)

Line 99 oomycetes repond (typo)

Line 116 It should be “ventral grooves” as there are no dorsal grooves on the zoospore.

Line 172 Please give examples of oomycete MAMPs and reference.

Line 288-295, The CRN-Avr3a system used by Schornack et al. [63] needs to be explained. [73] is a review paper, so the original publication [63] has to be sited.  The fusion protein experiment showed that LFLAK motif was needed for translocation, and the result does not contradict [63].  Please read [73] and [63] carefully.

Line 310 Pi04314, Pi03192 …respectively. Please verify the order of effectors and their subcellular localization.  According to Fig. 2, for instance, Pi03192 localizes to ER, not cytoplasm.

Line 341-343.  It is not clear which effectors inhibit ROS-mediated defense response.

Line 343 phytochrome-associated signaling.  SA, JA, ET .. are phytohormones, not phytochrome.

Line 367 (typo) can hijack this process

Line 387 “It is believed that Phytophthora species kill their host after successful completion …” (comment) P. infestans may kill its host, but not all Phytophthora species kill host at the time of sporulation.  Some species live on perennial hosts for a long time like a chronic disease.  The authors should not generalize from the P. infestans-potato interactions to all Phytophthora species.

Author Response

Reviewer #1

a)All authors are encouraged to read references more in-depth and also verify every gene and symbol in Figures and Tables carefully

The authors have ensured that all the references cited match the information presented. Also, we have corrected names of genes and symbols that were misspelled as seen below; 

b)In Table 2, soybean is one word.  P. sojae can infect several Lupinus species, but soybean is the only economically important host.  This indicates that “host range” can be artificial.

Here, the authors have included Lupin as one of the P. sojae host plants. Also we have spelled Soybean corrected (Table 1)

 c)Directions of suppression arrows in Fig. 1 are all wrong.  For instance, the arrow indicates as if HbSPA suppresses EPI10, but it is the other way around.  Please also enlarge arrowheads to make them easier to see.

The authors have enlarged arrowheads. The direction of the arrows has been changed accordingly. This was done to demonstrate effector-protease inhibition

d)Figure 1.  PIP1 not described in the text. GmGIP1, GmAP1: gene names should be consistent. EPIC2B, missing a letter. 

The authors have included protease PIP1 in the text (line 181). EPC2B has been corrected to EPIC2B.  Genes GmGIP1 and GmAP1 are two different genes however, their consistency has been maintained

e)Line 86 asexual, abd were (typo) and Line 99 oomycetes repond (typo)

The authors have corrected these typos lines 87 and 101

f) Line 116 It should be “ventral grooves” as there are no dorsal grooves on the zoospore.

The authors have corrected dorsal grooves to ventral grooves (Line 117)

g)Line 172 Please give examples of oomycete MAMPs

The authors have added examples of MAMPs (INF1, pep13 and CBEL) as shown in line 175-176:

h)Line 288-295, The CRN-Avr3a system used by Schornack et al. [63] needs to be explained. [73] is a review paper, so the original publication [63] has to be sited.  The fusion protein experiment showed that LFLAK motif was needed for translocation, and the result does not contradict [63].  Please read [73] and [63] carefully.

The authors admit that this was an unintentional mistake. We have therefore corrected to show that LFLAK motif is indeed crucial in effector translocation into the host cell as reported in a review paper (73). earlier on,  an experiment by (63) was conducted to demonstrate the significance of the motif in CRNs translocation . The two studies are in harmony ….line 294-296.

i)Line 310 Pi04314, Pi03192 …respectively. Please verify the order of effectors and their subcellular localization.  According to Fig. 2, for instance, Pi03192 localizes to ER, not cytoplasm.

The authors have corrected this to show that effector Pi03192 localizes to the ER… line 316

j)Line 341-343.  It is not clear which effectors inhibit ROS-mediated defense response.

We have clearly pinpointed effectors RxLR48 and Avr3b to inhibit ROS-mediated defense response (line 436 and 437)

k) Line 343 phytochrome-associated signaling.  SA, JA, ET .. are phytohormones, not phytochrome:Line 367 (typo) can hijack this process

We have corrected these typos

l) Line 387 “It is believed that Phytophthora species kill their host after successful completion …” (comment) P. infestans may kill its host, but not all Phytophthora species kill host at the time of sporulation.  Some species live on perennial hosts for a long time like a chronic disease.  The authors should not generalize from the P. infestans-potato interactions to all Phytophthora species.

The host-killing ability of Phytophthora species has been left open as suggested by the reviewer (Line 392-394). This is so because not all species of Phytophthora have been shown to kill their host to infect healthy plants  

Reviewer 2 Report

This manuscript presents a review on plant pathogenic genus Phytophthora. I definitely recommend its publication in this journal, however, I have some comments and corrections that I am sure will improve the manuscript even further

Author Response

Reviewer #2

  1. Is this apply for all?? (Figure 1)

Yes. Figure 1 describes a general mode of operation in Phytophthora spp (both below and above ground species).

  1. Typo in line 87

The authors have corrected this; from abd…to and

Reviewer 3 Report

The review is well researched and organized and covers the most important new discoveries of pathogenesis factors in the genus Phytophthora. In general, the manuscript is well written, however the English has to be checked for a number of grammar and spelling mistakes;

e.g. Lines 30-31: Of the three biotic factors, phytopathogens are by far a grave threat…

Line 50: ….spent every year to mitigate associated diseases but very little success.

Line 79: The dispersal phase is initiated by multinucleate sporangium…

Line 86: Zoospores of Phytophthora species are asexual, abd…

Line 89: ….followed by rapid cyst.

Many more mistakes were found throughout the manuscript.

Line 73: Not every Phytophthora is soilborne, therefore the sentence could be expanded to comprise all species. E.g.: In the presence of sufficient water in the soil or on the plant surface…

Line 234: To-date, genomes of several (not most) Phytophthoras species have been sequenced…

Line 257: please add genus name Plasmodium

Line 343 ff: Several plant pathogenic ‘true fungi’ produce hormones which are identical or at least very similar to plant hormones and play a role in pathogenesis (e.g. Fusarium). Maybe a sentence about synthesis or absence of plant hormones in phytophthora could be added.

Line 432: The authors mention the presence of viruses in several viruses, without discussing their potential role in pathogenesis. Viruses can play a very important role in regulating virulence in ‘true fungi’ (e.g. Cryphonectria parasitica, causal agent of chestnut blight) and are even used in biological control. The potential role of viruses in Phytophthora virulence should be discussed briefly.

Please make sure that every acronym is explained or written out when used for the first time in the manuscript (e.g. line 206: HR).

Figure 2: The meaning of the arrows in the figure should be explained (inducing vs. inhibiting)

Author Response

Reviewer #3

a) English has to be checked for a number of grammar and spelling mistakes;

As suggested, we have corrected this. For instance; typos and grammar in lines 32, 51, 81, 88, 91 and 101 have been corrected accordingly

b)Line 73: Not every Phytophthora is soilborne, therefore the sentence could be expanded to comprise all species. E.g.: In the presence of sufficient water in the soil or on the plant surface…

 We have responded to this by including leaf surface as a potential point of entry for above ground Phytophthora spp (line 74).

c)Line 234: To-date, genomes of several (not most) Phytophthoras species have been sequenced

We have corrected this (line 238)

d) Line 257: please add genus name Plasmodium

This has been corrected (line 262)

e) Line 343 f: Several plant pathogenic ‘true fungi’ produce hormones which are identical or at least very similar to plant hormones and play a role in pathogenesis (e.g. Fusarium). Maybe a sentence about synthesis or absence of plant hormones in phytophthora could be added.

The authors appreciate this suggestion. However, it is important to mention that introducing hormone synthesis in Phytophthora species at this point will totally change the facet of the strategy that is being discussed.  

f)Viruses can play a very important role in regulating virulence in ‘true fungi’ (e.g. Cryphonectria parasitica, causal agent of chestnut blight) and are even used in biological control. The potential role of viruses in Phytophthora virulence should be discussed briefly.

Line 440: Potential mechanism employed by viruses in enhancing virulence and mainly sporulation in Phytophthora spp is by inhibiting ammonium and amino acid intake.

g) Please make sure that every acronym is explained or written out when used for the first time in the manuscript (e.g. line 206: HR).

The authors have corrected this accordingly

h) Figure 2: The meaning of the arrows in the figure should be explained (inducing vs. inhibiting)

 Symbols used mainly arrows have been explained: The arrows in figure 1 indicate effector-protease inhibition while in figure 2 denote effector-target interaction and effector subcellular relocalization